# Implementation of COVID-19 Infection Control Measures by German Dentists: A Qualitative Study to Identify Enablers and Barriers

**DOI:** 10.3390/ijerph18115710

**Published:** 2021-05-26

**Authors:** Anne Müller, Florentina Sophie Melzow, Gerd Göstemeyer, Sebastian Paris, Falk Schwendicke

**Affiliations:** 1Department of Oral Diagnostics, Digital Health and Health Services Research, Charité-Universitätsmedizin Berlin, 14197 Berlin, Germany; anne.mueller@charite.de; 2Department of Operative and Preventive Dentistry, Charité-Universitätsmedizin Berlin, 14197 Berlin, Germany; florentina.melzow@charite.de (F.S.M.); gerd.goestemeyer@charite.de (G.G.); sebastian.paris@charite.de (S.P.)

**Keywords:** acceptance, COVID-19, interviews, qualitative, theoretical domains framework, SARS-CoV-2

## Abstract

Objectives: COVID-19 infection control measures have been recommended for dental practices worldwide. This qualitative study explored barriers and enablers for the implementation of these measures in German dental practices. Methods: Semi-structured phone interviews were conducted in November/December 2020 (purposive/snowball sampling). The Theoretical Domains Framework (TDF) and the Capabilities, Opportunities and Motivations influencing Behaviors model (COM-B) were used to guide interviews. Mayring’s content analysis was employed to analyze interviews. Results: All dentists (28–71 years, 4/8 female/male) had implemented infection control measures. Measures most frequently not adopted were FFP2 masks, face shields (impractical), the rotation of teams (insufficient staffing) and the avoidance of aerosol-generating treatments. Dentists with personal COVID-19 experience or those seeing themselves as a role model were more eager to adopt measures. We identified 34 enablers and 20 barriers. Major barriers were the lack of knowledge, guidelines and recommendations as well as limited availability and high costs of equipment. Pressure by staff and patients to ensure infection control was an enabler. Conclusions: Dentists are motivated to implement infection control measures, but lacking opportunities limited the adoption of certain measures. Policy makers and equipment manufacturers should address these points to increase the implementation of infection control measures against COVID-19 and potential future pandemics.

## 1. Introduction

In 2020, an unprecedented global pandemic, COVID-19, hit the world. COVID-19, the infection with SARS-CoV-2, was first described at the end of 2019 and has been tracked in Germany since 27 January 2020. The World Health Organization classified the spread of COVID-19 as a pandemic on 11 March 2020. The Robert Koch Institute (www.RKI.de; accessed on 26 January 2021) judged its risk to the German population as low to moderate in February 2020, and as high or—for risk groups—very high in March. Since these early days of the pandemic, governments have imposed various policies to contain the pandemic. Dental practices have been concomitantly subjected to various legislative and self-governing measures with the aim to reduce the risk of infection of patients, providers and staff. These measures focus on adjusting infrastructure and organization (plexiglass walls, air ventilation, longer appointments, staff rotations, reduction of high-risk activities) and personal protection equipment of staff and patients (PPE) [1]. The extent to which these measures were implemented, and their enforcement were highly variable between countries. In many countries, they further differed among regions (partially as a result of the heterogenous COVID-19 incidences and risks) or legislations (e.g., federal states, self-governing bodies).

A wide range of studies investigated the knowledge, attitudes and in some cases adoption of COVID-19 infection control measures (COVID-19 ICM) in dental practice. While the knowledge on COVID-19 has increased over the course of the pandemic, the majority of dentists have not yet implemented all recommended measures. A few dentists aimed to exceed recommendations or legal minimum standards [2,3,4,5,6,7,8,9,10,11,12,13,14,15,16,17,18,19,20,21]. It is, however, not clear why implementation differs between measures, dentists and settings. It is further not clear which measures were found to be sustainable over the pandemic.

Qualitative studies, for example using interviews, may foster an understanding towards the barriers and enablers of COVID-19 ICM, exploring in-depth factors facilitating or hampering the implementation in daily care. Qualitative research is also gaining popularity in the medical domain, mainly as it allows one to better understand reasons behind behaviors (i.e., going beyond describing them). Qualitative studies are fundamentally different from quantitative evaluations, as aspects of generalizability or the testing of any hypotheses are not relevant here; it is rather important to balance the need for in-depth, “rich” explorations of occurring themes with a wide representation of views. For example, Sandelowski [22] argues that sample sizes in qualitative studies should allow one to unfold a ‘new and richly textured understanding’ but should be small enough to enable ‘deep, case-oriented analysis’ [22] (p. 183). This balance leads to a rich and broad ensemble of themes and, eventually their saturation [23]. In interview studies, for example, saturation may occur after only a handful of interviews—depending on the question and analytic method employed—or may require a much higher number, especially for more complex and ambivalent topics [24]. Employing a systematic framework for such qualitative analysis allows for a comprehensive assessment of barriers and enablers. Further, the linkage with possible interventions on how to implement and sustain measures in the future becomes possible.

We aimed to explore barriers and enablers of COVID-19 ICM in German dental practices. We developed an interview guide based on the Theoretical Domains Framework (TDF) [25,26] and the Behavior Change Wheel (BCW) [27]. Semi-structured interviews with dentists were conducted to understand which capabilities, opportunities and motivations were relevant when wanting to alter their behavior towards ICM (COM-B model).

## 2. Methods

### 2.1. Qualitative Approach and Research Paradigm

This is a qualitative study; these studies do seldom come with representative samples and or sample size estimations; their aim is rather to generate a deep knowledge and sufficient variety to comprehensively reflect on possible relevant themes when concluding sampling. We developed an interview guide based on the Theoretical Domains Framework (TDF) [25,26] and the Behavior Change Wheel (BCW) [27].

Both the TDF and the BCW have been employed by a number of studies in dental re-search [28,29,30,31]. Combined, they present a paired framework in implementation science that allows linking capabilities, opportunities and motivations of stakeholders to identify barriers and enablers for changing behavior. The usage of BCW allows one to transparently and reproducibly link these behavior determinants to possible interventions that policy makers may want to employ to improve behavior change [27]. In our study, the behavior of interest was the adoption of COVID-19 ICM.

### 2.2. Research Characteristics and Reflexivity

Personal characteristics: A.M. conducted the interviews; A.M. has experience as an interviewer in dental implementation science from a range of previous studies with a background originally in biology. Interview transcripts of the first interviews were discussed among the whole research team to pilot the interview guide and possibly adapt it. The research team had in-depth discussions of relevant aspects, areas of interest, and tone after these first interviews.

### 2.3. Context

Twelve interviews were conducted with general dentists practicing in dental offices from all over Germany via telephone (Figure 1A). Interviews were carried out in the time period of mid-November to the beginning of December in 2020, a time where the COVID-19 incidence rates were highly dynamic in Germany. At this time, dentists (as well as the remaining German population) had experienced an early wave and a lockdown from March–April 2020, relaxations of restrictions and low incidences over summer as well as a second wave starting in October, with restrictions being re-enabled step-wise (from a soft lockdown in November to a full lockdown in December extending into 2021). Notably, this particular timing of our interviews and the associated setting-specific experiences with and incidence rates of COVID-19 may have affected dentists’ attitudes and behavior, which is why we display detailed information in Figure 1B–D.

### 2.4. Sampling Strategy

Purposive sampling was [32] used out of the research team’s network to identify possibly interested dentists across Germany (Figure 1A). The interviewed dentists needed to work in primary care and have minimum 2 years of experience in practice (e.g., were eligible to work or did already work in their own practice). We inquired only a minimal set of demographic variables (Table 1) to maximize anonymity and reduce barriers for participation. After the identification of interested dentists, snowball sampling was also used, i.e., our final sample consisted of dentists initially known to us and those unknown. Our main aim during sampling was a wide geographic and demo-graphic representation, as the implemented control measures but also the perceived risk of infection may have differed across Germany (given the highly heterogeneous incidence rates) observed, and also as we assumed gender or age to play a role in dentists’ attitudes towards the pandemic and the associated ICM.

There was no relationship established prior to the study. The dentists were informed about the study aim and encouraged to roam freely towards their observances and feelings in relation to COVID-19 in general and the implemented ICM.

In qualitative studies, as laid out, sampling aims for a broad representation of experiences and views to allow saturation of themes and a comprehensive understanding of behaviors of interest. Hence, purposive and snowball sampling are widely used. This is fundamentally different from quantitative survey studies, where sampling aims for representative samples allow for generalization to the target population [23]. Qualitative studies come, as laid out, with much smaller, “non-representative” samples, with the lower sample size being necessitated by the detailed, labor intensive data generation process involved.

We assessed the occurrence of new themes with each new interview, allowing us to evaluate if a saturation of themes occurred (Figure 2). Such saturation assessment has been proposed as a way of estimating the sample size for qualitative interviews [33]. As we found the last three interviews to yield only a few new themes, we concluded sampling and interviewing after the 12 interviews (Figure 3).

Several interviews were carried out with practices within one large city (D4, D8–D9, D12) due to very heterogenous incidences in boroughs.

### 2.5. Ethical Issues Pertaining to Human Subjects

Following written (e-mail) and verbal information about the study, the interviewees signed a declaration of consent. No financial incentive for participation was provided to the interviewees.

Ethical approval for the study was obtained from the Ethical Committee of the Charité (AZ: EA4/116/20). The reporting of the results follows the SRQR (Standards for Reporting Qualitative Research) checklist (Appendix A) [34].

### 2.6. Data Collection Methods

Twelve dentists, practicing in dental offices were interviewed in the time period of mid-November to the beginning of December in 2020. Interview guides were developed using the TDF and interviews were conducted in a semi-structured manner, allowing interviewees to elaborate in-depth and for new topics to emerge. Interviews were conducted early in the morning before the beginning of their clinical day, during lunch break or in the evening after work. Interviewees were assured that the content of the interviews would not be traced back to them during publication.

### 2.7. Data Collection Instruments and Technologies

The interview guides were generated by A.M. and two further experts in dental implementation science (G.G., F.S.) as well as further dental researchers (F.M., S.P.). Our interview guide covered the various domains of the TDF: (1) knowledge, (2) skills, (3) social influence, (4) social role, (5) environmental context and resources, (6) beliefs about capabilities, (7) beliefs about consequences, (8) reinforcement, (9) emotions, (10) memory attention and decision process, (11) optimism, (12) context and resources, (13) goals, (14) behavioral regulators, (15) perspectives and (16) social and professional identity. Most domains were reflected by more than one question, with considerable overlap between them, i.e., respondents’ answers may or may not have touched all possible domains associated with a specific question, allowing freedom for the in-depth exploration of subjectively relevant topics.

As discussed, the interview approach was piloted and adapted after the initial interviews and discussion among the team to yield insights into the capabilities, opportunities and motivations of dentists driving their behavior towards COVID-19 ICM. No repeat interviews were carried out. All interviews were conducted in German. The interviews were recorded using a voice-recorder; no field notes were taken during the interviews as only one interviewer (A.M.) was present. The interviews averaged 25 min per participant.

### 2.8. Data Processing, Data Analysis and Techniques to Enhance Trustworthiness

Interviews were anonymized. A.M. transcribed all interviews using f4 (v2 for Windows, dressing and pehl, Marburg, Germany). No transcription was returned to the participants.

The coding tree was developed based on the TDF domains. Inductive and deductive content analysis was conducted using Mayring’s principles [35]. Identified themes were classified as barriers, enablers or conflicting themes. The coding tree and the coding was double-checked by F.S. and G.G. No feedback was provided by participants towards themes and classifications. Themes and quotes were translated to English for publication by A.M. and double-checked by back translation once more by F.S. Themes were also mapped along specific COVID-19 ICM to allow an overview of how different measures were implemented and what barriers or enablers were relevant for implementation.

## 3. Results

According to O´Sullivan et al., 2020, we tried to structure especially the following results section in a very clear way [36] to enhance clarity and the reader accessibility of our findings. The twelve dentists (D1–D12, aged 28–71 years, four females and eight males) were located in towns with population sizes between 1633 (D11) and 3,762,456 (D4, D8, D9 and D12) inhabitants and spread across Germany (Figure 1A). Dentists were practice owners or employed (D4). Dentists worked alone (D11) or in group practices (e.g., D4 and D9). All except one (D9, an orthodontist) were general dentists. Further details as to the dentists, their practices, their organization and their experience with COVID-19 are shown in Table 1.

The majority of practices had implemented a range of ICM (Table 1). The measures which were most frequently not adopted were FFP2 masks (mainly as they were not comfortable or impaired treatment), face shields (as they were not applicable during treatments, i.e., impeded head bending), the organization in different teams (no practice could afford this given their staffing) or the avoidance of aerosol-generating treatments. Table 1 provides an overview about all measures, their implementation and barriers.

Figure 4 sums up barriers and enablers for each ICM. We identified 34 enablers and 20 barriers (14 themes were classified as conflicting) and organized them along the COM-B domains (Appendix B Table A1, with quotes). We briefly expand on them in the following section.

### 3.1. Knowledge

A major barrier in the early days of the pandemic was a lack of knowledge of adequate ICM; all interviewed dentists specified that information on ICM was provided too late from relevant stakeholders (Association of Dentist within the Statutory Insurance (KZV), Federal and Regional Dental Associations, Ministries). The absence of this information and a clear guideline (providing defined measures, with specific examples of mask types, shield manufacturers, etc.) was the strongest barrier at that time. Two dentists (D1 and D2) actively but unsuccessfully inquired about such information from their local bodies.

Eleven of the twelve dentists indicated that they obtained their knowledge about adequate ICM via active online search. Web pages of the Robert Koch Institute, the Swiss Dental Association, the Austrian Dental Chamber, Colloquio, Medscape and informative pages from Wuhan and the USA were visited in addition to web pages of German Dental Associations or KZV to obtain suitable information. Moreover, dentists actively exchanged knowledge with each other. While this exchange enabled some to implement certain measures, others found the general helplessness devastating and it increased their uncertainty (which was rather a barrier for further activity).


*D5: “Each Dental Association initially talked about something. There were no uniform recommendations from the Federal Dental Association”.*


Five of twelve dentists (D4–D6, D10 and D12) explicitly demanded a guideline in which exact ICM and also product recommendations could be found. An official guideline would also give more certainty how to deal with positive COVID-19 cases in the practice.


*D10: “I would have wished that a Dental Association /, because when I think about older colleagues who do not have PubMed or any contact /, how are they there? A kind of recommendation checklist of the Dental Association, where very clear product recommendations are given /”*



*D12: “A guideline would have somehow secured you in the event of a COVID-19 infection in practice /, if then somehow a patient calls afterwards and says: ‘Yes, I tested positive and was in your practice last week.’ One would have felt much safer”.*


### 3.2. Availability and Costs

Protective wear was not available at the beginning of March, and generally availability, orderability and costs played a major role. Initially, exchange with other colleagues led to the belief that it was acceptable to continue working without specific PPE.


*D5: “Right at the beginning, I was not sure whether I could have worked with a normal surgical mask /, yes, but it calms you down when you hear other colleagues who say: ‘Yes, we don’t get anything either. We also do it like that for now, right.’”*


Many of the interviewed dentists had or still have problems in procuring adequate protective materials. Some thought protective materials would be supplied by their regional dental association, which was not the case (D1, D2 and D5), and/or were surprised that prices had significantly increased (D3).


*D1: “They (concerning regional dental association) wanted to deliver to us too, but that was not possible. They could hardly take care of us”.*



*D2: “How do I get the things? Why do mouthguards or gloves cost more now than before?”*



*D3: “Lately I have actually been using masks, but at some point, you simply cannot stem the price”.*


Practices without high liquidation opportunities (e.g., without many privately insured patients) felt discriminated by a regulation allowing one to claim additional hygiene costs only for private patients, but not for the majority of statutorily insured patients.


*D2: “And this lump sum from private patients for increased hygienic standard, of course, only covers a fraction of the expanses. It is only available for private patients and it has been reduced although the prices are rising”.*



*D5: “For the private patients you can charge a lump sum for increased hygiene costs at least. For the statutory health insurance patients that is not allowed”.*


The fact that financial support was low led to some practices performing treatment as usual, mainly referencing economic pressure and disappointment towards the public attitude towards dentistry.


*D6: “/ if there are risk patients they will be informed in advance /, but otherwise /, since we were kicked out of the safety parachute /, because we were pushed into the hairdresser’s corner /, we do normal treatment”.*


### 3.3. Applicability of Measures during Treatment

Six of the twelve dentists regularly worked with FFP2 masks (D2, D5–D7, D9 and D12). Two of those additionally wore face shields (D6 and D12) and one additionally wore safety glasses (D2). Six of the interviewed dentists mainly wore surgical face masks during treatment (D1, D3, D4, D8, D10 and D11); two of them used safety glasses as well (D1 and D3). This was mainly grounded in FFP2 masks and face shields not being found applicable during treatment, e.g., impairing bending the head down.


*D1: “So I have to be completely honest about that. I just wear the normal surgical face mask and safety glasses of course. When you extract teeth, and you always hit your own rib cage with the face shield /, very difficult. I really have problems to work well with these things”.*



*D3: “We had face shields, but we do not use them because the glasses simply fog up”.*



*D7: “We all wear FFP2 masks now and we actually have to take them off every now and then because otherwise we get a headache. We have face shields. Personally, I do not wear them because I still have magnifying glasses. I did that at the beginning, but that is not my thing /. Because the magnifying glasses press completely on my eyes when the face shield is still in front of them. Then I cannot work properly”.*



*D10: “But I cannot /, I would suffocate if I had to wear a FFP2 mask too. That is why I am only wearing surgical face mask at the moment /, and you can try to look through a face shield with the magnifying glasses”.*


### 3.4. Experience with COVID-19

Dentists who had personal experience with COVID-19 (e.g., cases in their family, staff; e.g., D2 and D9) were generally more eager to adopt protective measures.


*D2: “So we have already been through quarantine ourselves and we only wear FFP2 masks, right down to the cleaner”.*


A clear connection between the size of population, incidence rates and degree of implemented ICM could not be seen (Table 1). In Berlin, two of four interviewed dentists implemented high standards (D9 and D12) while two others (D4 and D8) implemented only limited ICM. Indeed, the highest hygienic standard and ICM implementation was found in a very small town in Lower Saxony (D6). Here, the dentist advertised his practice, also on his webpage, as fulfilling the highest standards so that the fearful rural population was encouraged to come to the practice. It also became apparent that pressure by staff (also see next section) and patients to adopt these measures was an enabler.


*D6: “People get scared when someone accidentally leaves the treatment room without a mask. I have now noticed that one thing. We have to pay more attention to the fact that the people have to put back on their masks right after the treatment”.*


### 3.5. Impact of COVID-19 on Staff Availability

The impact of the pandemic on staff availability (staff being off sick with COVID-19 or quarantined due to symptoms; staff’s children being in home schooling) were mainly felt in major cities, where at the time of the interviews, incidence rates were highest (D4, D7 and D9).


*D4: “We have a Corona-denier in practice. I am an employee and can no longer enforce everything. That is conditioned from the practice owner. In my opinion, she is a little too indulgent”.*



*D7: “We had an employee who panicked in March and said she wants to quit the job /, because she is afraid of being infected”.*



*D9: “I think an employee was now on sick leave for 75 days /, easily for risk reasons by her doctor. Another assistant is now depressed, because she could not cope with the psychological stress of being at home”.*


Staff rotation for reducing cross-infection risks was felt impossible by most dentists.

## 4. Discussion

ICM for dental practices are highly relevant to allow sustained services provision during pandemics such as the one the world is experiencing. Given the high risk of exposure to potentially infectious liquids and aerosols during dental treatments, protecting the dentists, staff and other patients is highly relevant in this setting. A wide range of measures have been developed, adapted and implemented for COVID-19 infection control in dental practices over the course of the year 2020. While there have been differences in the specific recommended measures as well as their enforcement around the globe (from complete shutdowns of practices to a more liberal approach of individual decision making on practice level), it is relevant to understand why dentists have implemented or not implemented certain measures. The present study employed qualitative interviews and analyses along the TDF and the BCW to understand capabilities, opportunities and motivation shaping the implementation of ICM around COVID-19 in German dental practices. We identified more than 50 themes and a range of specific barriers and enablers, which could be used by decision makers to improve implementation or abstain from recommending efficacious, but not implementable measures. Our findings may, to some degree, be transferred to other settings and may generally inform pandemic preparedness packages for dental practices for the future.

A number of findings require detailed discussion. First, and different from many other analyses around behavior change in dental practice, a major barrier to implementation was a lack of knowledge and capability. Dentists were highly motivated to adopt measures allowing them to maintain their services and keep their practices open in the early days of the pandemic; they did not need to be convinced that ICM were useful. In many previous studies, it was shown that knowledge was not a major factor to enable behavior change, but that lacking motivation was the main barrier to change [28,29,30,31]. This finding is relevant, as knowledge transfer and capacitation are likely easier to achieve than inspiring motivation. Notably, the lack of knowledge focused mainly on the early phase of the pandemic, when a dramatic unpreparedness of many health services, including dental health services, for dealing with such pandemic became apparent. We showed that by now, limited access to knowledge has been addressed, and that factors such as capability and opportunity are becoming more important.

Second, this lack in options to do the right thing (lacking opportunity) seems the main barrier at the current phase of the pandemic. Limited access to equipment and materials or high costs have been identified as main barriers. Policy makers and professional bodies are called to action and to increase the accessibility of these measures. Dentists are highly motivated and by now knowledgeable and should not be limited in their action radius by lacking access. German dentists were especially disappointed that public policies to support health services did not consider dentistry, the reasons of which we cannot cover here. Financially supporting dental services, which can be seen as essential as other medical services, during the pandemic might be advisable given the expected economic impact of lockdowns, but also ICM on practices [37]. Specifically, dentists approved fee items allowing reimbursement for infection control equipment and materials, something that private insurers in Germany have granted, while the public insurance refused such support. The feeling of being left alone and not being acknowledged for working at a high-risk zone for infection in an essential health service was a major barrier for dentists to embrace (costly) ICM.

Third, the applicability of certain, theoretically efficacious measures was criticized. Specifically, face shields were near-unambiguously found as unsuitable to wear during treatment, impeding head mobility and bending down, for instance. In addition, FFP2 masks were claimed to be highly uncomfortable when worn throughout the whole day. Manufacturers should address such specific critiques.

Last, it became apparent that measures which may be applicable in larger clinics, such as adjusting staff rotations or concentrating specific high-risk procedures around specific staff members, times during the day or zones in the practice, were not implementable in single or small-group practices, which are typical for German dental care. The full centralization of managing COVID-19-positive patients in specific “COVID-19 practices”, as has been the case in Germany since summer, seems to be a sensible measure in this regard. Centralizing services in such way and thereby concentrating risks of exposure should be considered for further groups or procedures if needed.

This study comes with a range of strengths and limitations. First, it yielded detailed data on dentists’ reflections on the pandemic in general and the recommended ICM. The resulting insights allow a deep understanding of barriers and enablers to implementation both on an emotional but also a factual level. Notably, our findings can be triangulated with those of quantitative evaluations (e.g., from surveys), which highlighted that knowledge on COVID-19 increased with time and was generally not the main barrier for ICM in more recent surveys; that the adoption of PPE varied widely and that some PPE, such as face shields and FFP2-masks, were not implemented regularly; that there was uncertainty regarding how to adapt practice infrastructure and organization to reduce infection risks and that, generally, the specific setting was associated with the implementation of measures [2,3,4,5,6,7,8,9,10,11,12,13,14,15,16,17,18,19,20,21]. Our study now adds insights as to why all these observations were made. Second, and as a limitation, our sample was small and as usual for qualitative studies, not representative. Our findings will not be fully generalizable to all dentists in Germany and may not fully transfer to other healthcare systems (where other measures are recommended and opportunities may differ). In addition, we interviewed dentists at a certain time point of the pandemic. Our findings are hence a snapshot and interviews during another phase of the pandemic may yield different insights. It is relevant to interpret our findings in this light and to link them with the incidence rates shown in Figure 1; since over the winter of 2020/21, incidence rates have surged and experiences with COVID-19 will be more common among dentists by now. Interviews now or when vaccination has been more widely performed will yield different insights. Additionally, the importance of the specific operational environment was highlighted by our study, as dentists from practices in large cities (with higher incidence rates at the time of the interviews) cited other barriers and enablers than those from rural areas (where there were only few cases at this time and nearly no personal experience with COVID-19). Third, and as mentioned, our study may, given its qualitative character, not claim representativeness of the identified barriers and enablers. Due to our sampling strategy, we can just give insights in the situation of a few cities in Germany. Here, a saturation of themes was reached but it should be mentioned that it could be again another situation for dentists in Bavaria, for example. Additionally, using the TDF allows for comprehensive and systematic interviewing and analysis, and linking the barriers and enablers to specific measures further allows one to exhaustively assess their implementability. Using the TDF further allows the linkage of themes with the BCW, something that was beyond our scope here but could be considered in the future.

## 5. Conclusions

In conclusion, the interviewed dentists were highly motivated to implement ICM, but a lack of knowledge and opportunities limited the adoption of certain efficacious measures. Certain ICM were found to be inapplicable during dental treatment or in dental practices, and specific equipment remained unavailable or generated high costs. Policy makers and equipment manufacturers should address these points to increase the implementation of ICM against COVID-19.

## Figures and Tables

**Figure 1 ijerph-18-05710-f001:**
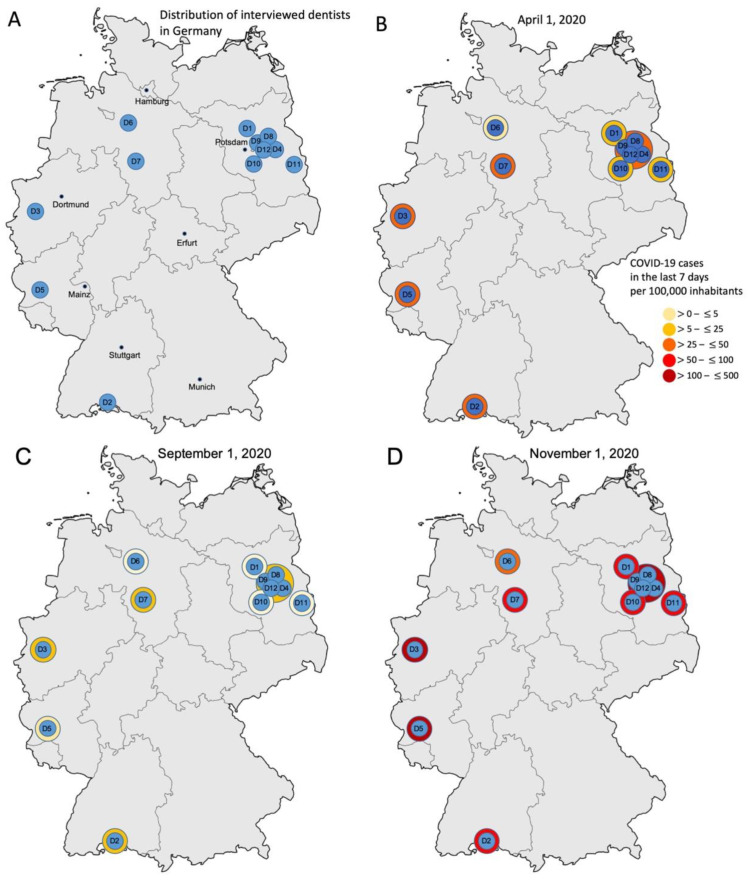
Distribution of interviewed dentists (D1 to D12) in Germany and COVID-19 incidence. (**A**) Distribution of dentists D1–D12. The concrete location of dentists and practices is anonymized, but some German cities are listed by name for orientation. (**B**–**D**) Incidence rates of COVID-19 at three different time points in 2020 (cumulative cases over the last 7 days per 100,000 inhabitants), according to the official body of the German health surveillance, the Robert Koch Institute (www.rki.de; accessed on 26 January 2021).

**Figure 2 ijerph-18-05710-f002:**
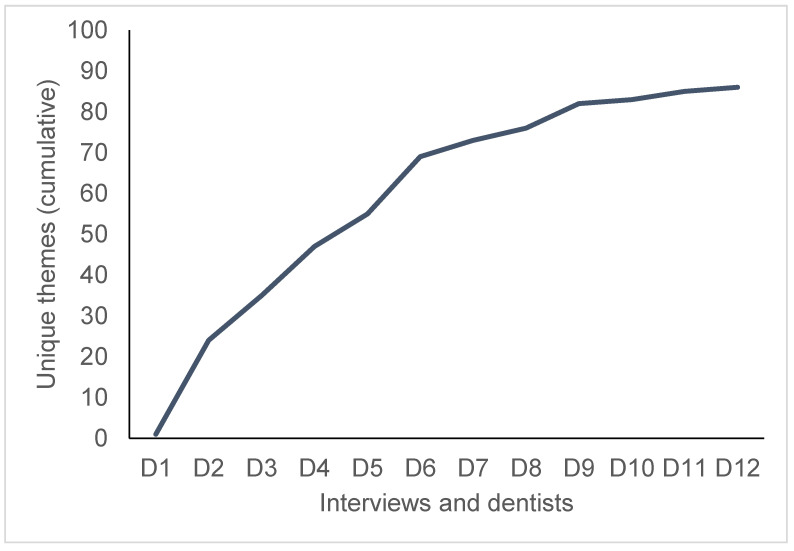
Saturation of themes. Twelve interviewed dentists (D1–12).

**Figure 3 ijerph-18-05710-f003:**
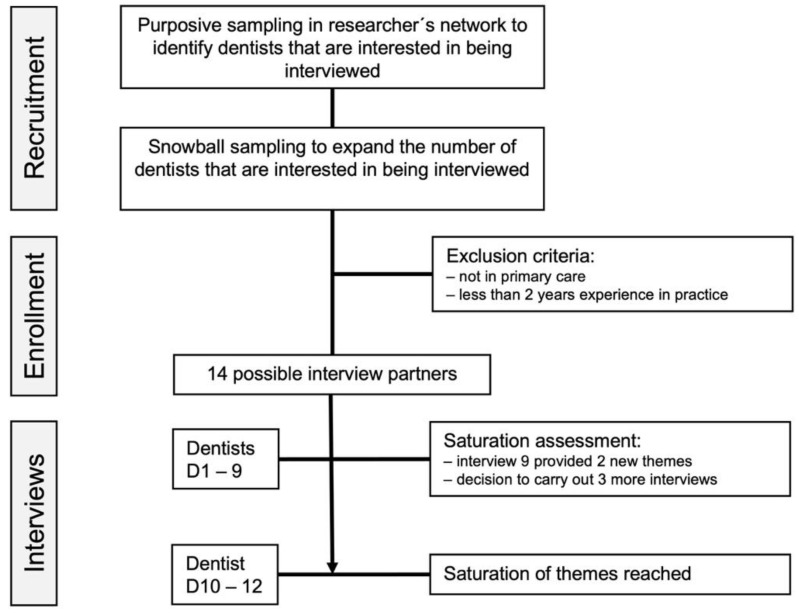
Sampling strategy.

**Figure 4 ijerph-18-05710-f004:**
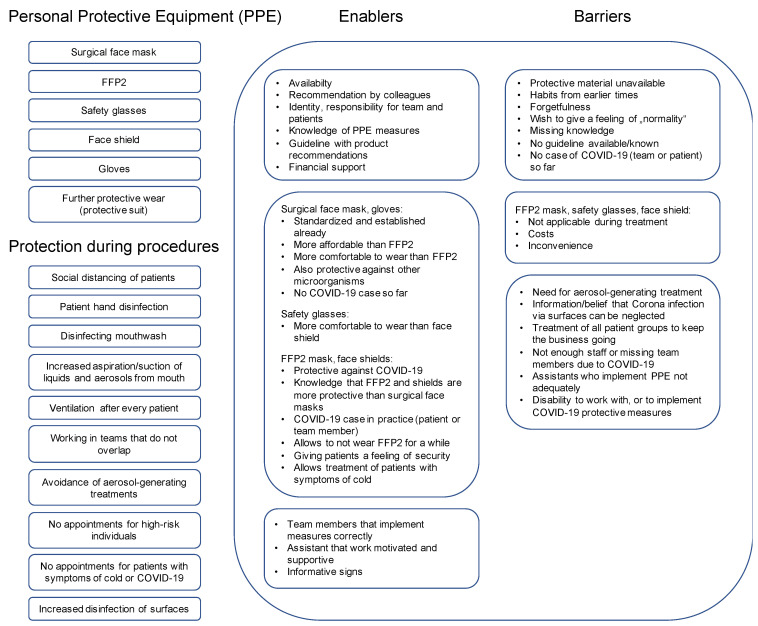
COVID-19 infection control measures and associated enablers and barriers. The infection control measures are subdivided into personal protective equipment (PPE) and protective measures during working procedures in dental practice (**left**). Enablers and barriers for the implementation of these measures are shown in the right boxes. The first sub-box provides general enablers or barriers. The second sub-box refers enablers and barriers to PPE and the third sub-box to enablers and barriers to protection during procedures.

**Table 1 ijerph-18-05710-t001:** Dentists’ (D1 to D12) and practices’ characteristics (upper panel) and the implementation of different infection control measures for COVID-19 (lower panels).

Dentist (Gender, Age at 10.12.20)	D1 (f, 43y)	D2 (f, 41y)	D3 (m, 60y)	D4 (m, 71y)	D5 (m, 37y)	D6 (m, 48y)	D7 (f, 45y)	D8 (m, 67y)	D9 (m, 35y)	D10 (m, 47y)	D11 (m, 49y)	D12 (f, 28y)
Location of dental surgery (federal state of Germany)	Brandenburg	Baden-Wuerttemberg	North Rhine-Westphalia	Berlin	Rhineland-Palatinate	Lower Saxony	Lower Saxony	Berlin	Berlin	Brandenburg	Brandenburg	Berlin
Population of town	31,000	85,524	619,294	3,762,456	18,762	3838	532,163	3,762,456	3,762,456	13,984	1633	3,762,456
COVID-19 quick tests available?	Yes	No	No (ordered)	No	Yes	Yes	No	Yes	Yes	No	No	Yes
COVID-19 quick test used?	In case of symptoms	n/a	n/a	n/a	In case of symptoms	Regularly	n/a	Regularly	In case of symptoms			In case of symptoms
COVID-19 + tested staff?	No	Yes	No	No	No	No	No	No	Yes	No	No	No
COVID-19 + patients in the past?	No	No	No	No	Yes	Yes	No	No	No	No	No	Yes
Staff with high-risk individuals?	No	No	No	Yes	Yes	No	No	Yes	Yes	No	No	n/a
Number of patients	Increased	Decreased	Decreased	Unchanged	Decreased	Decreased	Decreased	Decreased	Decreased	Decreased	Unchanged	Decreased
Interaction on infection control with colleagues	Yes	Yes	Yes	Yes	Yes	No	Yes	Yes	Yes	Yes	No	Yes
Personal protective equipment (PPE) dentist, staff												
Mask (type) between treatments	Surgical face mask	FFP2	Surgical face mask	No mask (standard) or surgical face mask	FFP2	Surgical face mask	FFP2	No mask	FFP2	Surgical face mask	Surgical face mask	Surgical face mask or FFP2
Mask (type) during treatment; surgical face mask, FFP2	Surgical face mask (standard)	Not mentioned	Surgical face mask	Surgical face mask	Not mentioned	Not mentioned	Not mentioned	Surgical face mask	Not mentioned	Surgical face mask	Surgical face mask	Not mentioned
	FFP2 not applicable during treatment	FFP2	FFP2 is too expansive	FFP2 sometimes	FFP2	FFP2	FFP2	FFP2 is too uncomfortable	FFP2	FFP2 not applicable during treatment	FFP2 not applicable during treatment	FFP2
Gloves	Yes (standard)	Yes	Yes	Yes	Yes	Yes	Yes	Yes	Yes, disinfected before use	Yes	Yes	Yes
Safety glasses or face shield during treatment	Safety glasses	Safety glasses	Safety glasses	Not mentioned	Not mentioned	Not mentioned	Not mentioned	Not mentioned	Not mentioned	Not mentioned	Not mentioned	Not mentioned
	Face shield is not applicable during treatments	Face shield is not applicable during treatments	Face shield is not applicable during treatments	Face shield optional	Face shield is not applicable during treatments	Face shield	Face shield is not applicable during treatments	Face shield is too uncomfortable	Face shield optional	Face shield optional	Face shield is not applicable during treatments	Face shield
Protective suit in addition to workwear (during treatment)	No (standard)	No	No	No	No	Yes	No	No	No	No	No	No
Different PPE for staff	Not mentioned	Not mentioned	Yes, face shield for prophylaxis	Yes, decided individually	Yes, face shields	No	Yes, face shields	Yes, FFP2 for prophylaxis	No	Yes, FFP2 and surgical face	Not mentioned	No
General measures												
Social distancing during breaks	Yes	Yes	Yes	No	Yes	Yes	Yes	No	Yes	Yes	No	Not mentioned
More frequent surface disinfection	Yes	No	Yes	Not any longer	Yes	Yes, UV-lamps	Yes	Yes	Yes	Yes	Yes	Yes
Adequate hand hygiene	Yes	Yes	Yes	Yes	Yes	Yes	Yes	Yes	Yes	Yes	Yes	Yes
Problems with team members due to COVID-19?	No	No	No	Yes	No	No	Yes	No	Yes	No	No	No
Organisation—Appointments (apmt.) and patients												
Apmt. for health patients	Yes	Yes	Yes	Yes	Yes	Yes	Yes	Yes	Yes	Yes	Yes	Yes
Apmt. for emergencies without symptoms of cold	Yes	No	No	No	Yes	Yes	No	Yes	No	Yes	Yes	Not mentioned
Apmt. for patients with symptoms of cold	No	No	No	No	No	Yes	No	No	No	No	Yes	No
Apmt. for patients from high-risk group	Yes (a lot)	Yes (a lot)	Yes	Yes	Yes	Yes	Yes	Yes	No	Yes	Yes	Not mentioned
Apmt. for COVID-19 positive patients	No	No	No	No	No	No	No	No	No	No	Not clear	No
Prolonged appointments	No (standard)	Yes	Yes	No	Yes	Yes	No	Yes	Yes	No	No	Yes
Hand disinfection (entry/leave)	Yes/optional	Yes/optional	Yes/optional	Yes/optional	Yes/optional	Yes/optional	Yes/optional	Yes/optional	Yes/optional	Yes/optional	Yes/optional	Yes/optional
No physical greeting	Yes	Yes	Yes	Yes	Yes	Yes	Yes	Yes	Yes	Yes	Yes	Yes
Face mask constant (except during treatment)	Yes	Yes	Yes	Yes	Yes	Yes	Yes	Yes	Yes	Yes	Yes	Yes
Inquiry about COVID-19 symptoms	Yes	Yes	Yes	Yes	Yes	Yes	No	No	No	No	No	Yes
Temperature measurement	No	No	No	No	No	No	No	No	No	No	No	No
Planned patient flow	Yes	Yes	Yes	Sometimes not	Yes	Yes	Yes	Yes	Yes	Yes	Yes	Yes
Waiting room												
Information signs for safe behavior	Yes	Yes	Yes	Yes	Yes	Yes	Yes	Yes	Yes	Yes	Yes	Yes
Restricted number of patients and distance	Yes	Yes	Yes	Sometimes not	Yes	Yes	Yes	Yes	Yes	Yes	Yes	Yes
Restricted access to toilet	No (standard)	No	No	No	No	Yes	No	No	No	No	No	No
Accessoires present (magazines…)	No	No	Magazines	No	No	No	Magazines	Yes	No	No	Yes	Not mentioned
Treatment												
Different teams	No	No	No	No	No	No	No	No	No	Yes (standard)	n/a (single dentist)	Not mentioned
Avoidance of certain treatments	No	No	Yes, x-ray from outside the mouth prefered to avoid coughing	Yes, reduced prophylaxis	No	Yes, reduced prophylaxis	Yes, reduced prophylaxis	Yes, reduced prophylaxis	Yes, no prophylaxis	No	No	No
Disinfecting mouthwash	No (H_2_O_2_ was tried to use but foam blocked units)	Yes, Cool Mint from Listerine or chlorhexidine digluconate (acc. to recent information, also H_2_O_2_ was not orderable)	Yes, H_2_O_2_	Yes, chlorhexidine digluconate (acc. to recent information, previously H_2_O_2_)	No (dentist has no information)	No longer (information that chlorhexidine digluconate is ineffective)	Yes, H_2_O_2_	Yes, H_2_O_2_	Yes, H_2_O_2_	Yes, Betaisodona or Cool Mint from Listerine	Yes, chlorhexidine digluconate (standard)	Yes (no further information)
Ventilation after every patient	Yes (standard)	Yes	Yes	Yes (standard)	Yes	Yes, also air filter	Yes	Yes	Yes, also air filter	Yes (standard)	Yes (standard)	Yes
Aspiration of liquids from mouth	No (standard, patient spits out)	No	Aspiration preferred	No	No	Aspiration prefered	Aspiration (standard)	No	No	Aspiration (standard)	No	No
Changing of face protection	After each patient	At start of work	After each patient	Rarely, is often forgotten	Every 4 h	Daily	Daily	Approximately after 5–6 patients	Every 2 h	After each patient	After each patient	Not mentioned
Infrastructural measures	No	Plexiglass at reception	Plexiglass at reception	Plexiglass at reception	Plexiglass at reception	Plexiglass at reception	Plexiglass at reception	No	Plexiglass at reception, protective walls	Plexiglass at reception	Plexiglass at reception	Plexiglass at reception

Implemented measures are shown in green, those which were not implemented in red, with identified barriers (if mentioned). White boxes indicate measures which had been in place already independently from COVID-19 infection control. FFP2, Filtering Face Piece with protection class 2; Apmt, Appointment.

## Data Availability

Data used in this study cannot be made available by the authors given data protection rules.

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
