# Peer review of "Implementation of COVID-19 Infection Control Measures by German Dentists: A Qualitative Study to Identify Enablers and Barriers"

_ijerph, 2021, doi:10.3390/ijerph18115710_

Round 1

Reviewer 1 Report

the authors have modified all the questions requested.

Author Response

Many thanks for your review. We are glad to have responded to all your questions.

Reviewer 2 Report

Thank you for the opportunity to review the manuscript entitled "Implementation of COVID-19 infection control measures by German dentists: A qualitative study to identify enablers and barriers".

As I indicated in my previous review I consider the sample size used by the authors to be very insufficient. They only evaluated the questionnaire of 12 persons.

According to the previous review the authors do not extend the sample size.

In this new modified manuscript the authors, for example, do not adequately describe the bibliographical references according to the journal's guidelines.

The authors have not improved the manuscript according to the previous recommendations.

In my opinion this article cannot be accepted for publication.

Author Response

We have extensively explained why the sample size is justified and appropriate. With all due respect we would also like to ask you to read the manuscript properly: We have NOT used a questionnaire, this is a qualitative interview study, where sample sizes of 12 are acceptable, as outlined. We cannot make more efforts in explaining why we arrived at this sample size, so we are sorry to not have satisfied your suggestions. We will once more check the references; note however, that they are formatted by the journal in parts.

Reviewer 3 Report

This is a well-written, well-conducted qualitative study of German dentists concerning enablers and barriers to the adoption of infection control measures during the COVID-19 pandemic. The paper is very complete and comprehensive and offers an extensive discussion of the rationale, the methods, and the results. The study used a theoretical framework to design the questionnaire. It might be useful if the authors included the interview guide as an appendix or within the manuscript as a table. The findings are interesting and can be used to inform policy surrounding infection control during a pandemic and can also be used by manufacturers of PPE to improve masks and face shields.

The English is good in most of the manuscript but could be edited for clarity, grammar and punctuation.

Some suggestions:

  • for readers not familiar with Germany, it would be useful to put some of the larger cities on the map in Figure 1 (Berlin, Munich, Frankfurt, Hamburg, Cologne)
  • also on the map, a title for the upper left map showing the distribution of dentists in the study would make it easier to understand at first glance
  • also, in terms of the map, might it be better to make the entire state a color according to covid rate rather than small circles? although this may be true as in Germany apparently covid cases were largely urban - maybe some explanation would be useful in the description under the table

The one issue is that it appears that there are large areas of Germany where there was no dentist included in the sampling - perhaps this is because the sampling strategy relied on dentists known by the authors. One wonders if dentists from Bavaria or the center of the country might have had a different set of opinions - although saturation was reached, it was reached only in those areas where dentists were sampled and some areas included more than one dentist. This is a minor issue, but should be noted in the limitationsl

Author Response

Many thanks for these useful recommendations, which we happily took up.

Round 2

Reviewer 2 Report

In my opinion the article is not suitable for publication.

Authors should take into account the opinion of the reviewers in order to improve their work.

This manuscript is a resubmission of an earlier submission. The following is a list of the peer review reports and author responses from that submission.

Round 1

Reviewer 1 Report

The study design is interesting, but the sample is very limited and the authors do not provide information on the calculation of the sample size necessary to obtain meaningful results that can be extrapolated to the entire population of German dentists.

Author Response

We have revised the paper according to your comments. Please let us highlight once more that this is a qualitative study, where representative sample sizes are not an objective, but saturation of themes. We conducted a separate and additional analysis on this and also placed a standard reference accepted in the field fo this purpose. Sample size estimations a priori are not performed in qualitative studies, while using the added methodology we can at least demonstrate that we had sufficient "power" (another concept not used in qualitative sciences) to comprehensively cover themes. We moreover once more highlight the sample size and the limitations of qualitative studies. 

Reviewer 2 Report

Dear Authors, sorry to say but for me it is not understandable that nearly 8 month after beginng of the pandemic still " a majority of dentists (in Germany) have not yet implemented all recommended measures." especially treatments that are not absolutel necessary are posponed and there were less consulttions per day. I could not find the publication by A. Amato et al. (2. July 2020) with an instructive patient flowchart screening for non-emergency dental care. I could not unterstand quite well the expressions: enablers and barriers. "pressure by staff and patients was an enabler"..,pave the way to get PPE? Concerning study design (107): how were the dentists selected, have you known them. Have you ask more and only 12 have answered. Snowballing sampling (110)? (12 are not representative but it was mentioned by yourself (367), app. 42 000 dentists in Germany).Overall it is a very good informative and instructive (especiallyfor the dentists reading the publication)  snapshot of a short period of the ICM situation in dental offices during the COVID-19 pandemic

Author Response

We have revised the paper according to your comments.

The description of enablers and barriers was clarified.

Sampling details and the issues of sample size have been addressed in line with other reviewer comments. Please let us highlight once more that this is a qualitative study, where representative sample sizes are not an objective, but saturation of themes. We conducted a separate and additional analysis on this and also placed a standard reference accepted in the field fo this purpose. 

Reviewer 3 Report

This scientific paper with a sample of 12 subjects cannot be considered suitable for publication.

The authors should consider increasing the sample in order to consider the manuscript for publication.

On line 366 the authors indicate that the sample is not representative.

Author Response

We have revised the paper according to your comments. Please let us highlight once more that this is a qualitative study, where representative sample sizes are not an objective, but saturation of themes. We conducted a separate and additional analysis on this and also placed a standard reference accepted in the field fo this purpose.We moreover once more highlight the sample size and the limitations of qualitative studies.

Round 2

Reviewer 1 Report

The authors have provided more info about the power of the design and have answered all the questions.

Author Response

Many thanks for your assistance in improving the paper.

Reviewer 3 Report

This work is not suitable for publication in this journal.

Authors should carefully review their work.

The authors should substantially expand the sample for their study.

Author Response

Many thanks for your review. We have outlined that sample size is not a criterion for a qualitative study, and demonstrated saturation of themes. We appreciate that under the view of quantitative evaluations - something we ourselves have done hundred of times - the sample size matters and this one would be problematic. For the scope of our study, however, this was sufficient, as demonstrated. We have revised the paper language wise once more. Many thanks.